# Identifiability Guarantees for Causal Disentanglement from Purely Observational Data

**Ryan Welch**\*
Massachusetts Institute of Technology
Broad Institute of MIT and Harvard

**Jiaqi Zhang**\*
LIDS, Massachusetts Institute of Technology
Broad Institute of MIT and Harvard

**Caroline Uhler**
LIDS, Massachusetts Institute of Technology
Broad Institute of MIT and Harvard

## Abstract

Causal disentanglement aims to learn about latent causal factors behind data, holding the promise to augment existing representation learning methods in terms of interpretability and extrapolation. Recent advances establish identifiability results assuming that interventions on (single) latent factors are available; however, it remains debatable whether such assumptions are reasonable due to the inherent nature of intervening on latent variables. Accordingly, we reconsider the fundamentals and ask what can be learned using just observational data.

We provide a precise characterization of latent factors that can be identified in nonlinear causal models with additive Gaussian noise and linear mixing, without any interventions or graphical restrictions. In particular, we show that the causal variables can be identified up to a *layer*-wise transformation and that further disentanglement is not possible. We transform these theoretical results into a practical algorithm consisting of solving a quadratic program over the score estimation of the observed data. We provide simulation results to support our theoretical guarantees and demonstrate that our algorithm can derive meaningful causal representations from purely observational data.

## 1 Introduction

Advances in representation learning play a pivotal role in the application of machine learning across various fields, including natural language processing, computer vision, and life sciences (c.f., [6, 52]). The emerging field of causal disentanglement holds the promise to augment such advances by identifying and learning some aspects about the latent causal factors behind data [39, 18]. These latent causal factors have been shown to improve the interpretability of high-level concepts behind complex high-dimensional data [23, 32, 60, 53] and enable extrapolation to predict how novel interventions will affect the data [57, 59, 36].

In pursuit of causal disentanglement, two *critical* questions are: (1) to theoretically understand to what extent the latent causal factors are identifiable, and (2) to algorithmically design efficient methods to learn these factors with finite samples. Despite the recent surge of interest in this area, these questions remain difficult given the inherent challenges of both disentanglement and causal discovery. In the disentanglement literature, the latent factors are assumed to be independent, and it is known that identifying them is not possible without further knowledge on the data-generating process [15]. Relaxing the independence assumption, causal disentanglement considers potentially related latent

---

\*Equal contributions.

38th Conference on Neural Information Processing Systems (NeurIPS 2024).

factors and aims to discover not only the latent factors but also their latent causal relations. Since this extends disentanglement, the latent factors are unidentifiable without additional information. Furthermore, learning causal relations is notoriously challenging as the number of variables grows: the underlying structure is generally not unique [4], and it is computationally and sample inefficient to learn complex causal graphs [12, 51].

To overcome these difficulties, a trend in recent works has been to consider having access to interventional data, where a common assumption is to assume that interventions on all single latent factors are available [43, 2, 47, 59, 50, 48, 9]. Although a goal of causal disentanglement is to be able to control individual latent factors, it is debatable whether assuming existence of direct interventions on latent factors is reasonable [36, 7, 49, 3], since one can argue that direct interventions on (single) latent factors make these factors non-latent. Furthermore, it might be infeasible to perform interventions on (some of) the factors due to ethical or cost reasons. As a consequence, it is important to understand what can be achieved solely based on observational data.

In our work, we consider causal disentanglement from purely observational data. The key idea behind our approach is to utilize *asymmetries* in the joint distribution of the latent factors. In particular, we consider latent factors that are generated by an unknown nonlinear causal model with additive Gaussian noises, from which we obtain observations after an unknown linear mixing. Nonlinear models with additive Gaussian noises have been a popular choice in the causal discovery literature due to their flexibility, intriguing identifiability properties (in the fully observed setting), and benign statistical sample complexities [31, 38, 35, 61]. These models imply asymmetric relationships between causes and effects, which can be utilized to distinguish causal directions. Beyond their theoretical properties, these models are commonly chosen to represent real world causal systems, such as gene regulatory networks [16], given their ability to fit non-parametric relationships.

**Contributions and Organization.** We define nonlinear additive Gaussian noise models in the context of causal disentanglement in Section 2. We provide a precise characterization of latent factors that can be identified in such models, with purely observational data and no graphical restrictions. In particular, we show that the latent variables can be identified up to a layer-wise transformation that is consistent with the underlying causal ordering, and that further disentanglement is not possible. These results are provided in Section 3. We transform these theoretical results into practical algorithms in Section 4 by building upon recent successes of combining score matching and causal discovery [35, 29] to devise a method that solves a quadratic program over the score estimation of the observed data. The resulting algorithm enjoys efficiency and flexibility to be combined with any existing off-the-shelf score estimation method. We demonstrate our results empirically with simulations in Section 5, and conclude with a discussion in Section 6.

## 1.1 Related Work

**Causal disentanglement.** Previous works in causal disentanglement have mostly considered varying assumptions on: the available data, the underlying causal model of the latent factors, and the mixing function between latent factors and observed data. Assuming the availability of interventional data, [24, 43, 9] established results for parametric causal models, whereas [47, 2, 59, 50, 17, 48] studied non-parametric causal models. Most of these works assume linear mixing (or a special case of polynomial mixing that can be easily reduced to linear functions), with the exception of [9, 50, 17, 48], where stronger assumptions on either the parametric causal model or more interventions are required to compensate for the general mixing functions. Prior to these works, [1, 8] established results assuming counterfactual data, which usually leads to stronger identifiability as one can now contrast counterfactual pairs.

Few recent works considered identifiability without interventions [45, 58, 56]. These works typically assume that (parts of) the latent factors can be observed after multiple different mixing functions. In the case where only one observational dataset is available, which is the setting of this paper, previous works have obtained results assuming both parametric models as well as additional structural restrictions on the mixing function [11, 19, 54, 55, 21]. Such structural restrictions refer to constraints on the set of latent variables that determine each observed variable, which is distinct from functional restrictions on the mixing function such as linearity. An example of such restrictions is the pure child assumption [40, 14], specifying that each observed variable has only one latent parent. To the best of our knowledge, *our work is the first to establish identifiability guarantees of causal disentanglement*

Table 1: **Comparison of our results to prior works on causal disentanglement.** For the latent model, *L* stands for linear mechanisms whereas *NL* stands for nonlinear mechanisms; *G* stands for Gaussian noise whereas *NG* stands for non-Gaussian noise; *Discrete* refers to discrete causal variables. Here, we summarize the identifiability results in terms of latent causal graph identification.

| | Data | Latent Model | Structural Mixing | Identifiability Results |
|---|---|---|---|---|
| [1, 8] | Counterfact. | General | **No** | Fully identifiable. |
| [43, 9] | Hard Interv. | LG | **No** | Fully identifiable. |
| [2, 49, 48, 50] | Hard Interv(s). | General | **No** | Fully identifiable. |
| [49, 59] | Soft Interv. | General | **No** | Up to transitive closure. |
| [45] | Multi-view | LG | **No** | Block-wise identifiable. |
| [58] | Multi-view | NL | **No** | Block-wise identifiable. |
| [14, 19] | **Purely Obs.** | Discrete | Yes | Up to Markov equivalence. |
| [11, 54, 55] | **Purely Obs.** | LNG | Yes | Fully identifiable. |
| [21] | **Purely Obs.** | NL | Yes | Fully identifiable. |
| **This work** | **Purely Obs.** | NLG | **No** | Up to causal layers. |

*in the purely observational setting without imposing any structural assumptions over the mixing function.* We summarize these comparisons to prior works in Table 1.

We additionally recognize that identifiability of latent factors from purely observational data has been considered outside of causal disentanglement [35, 20]. However, these results do not extend to the setting considered in this work, given the assumed data generating processes to do encapsulate the causal graph.

**Score matching in causal discovery.** Since our algorithm builds upon discovery methods using score matching, we briefly review these approaches. Works in this direction have mainly focused on causal discovery when all causal variables are observable in identifiable paramteric causal models such as nonlinear additive Gaussian noise or additive non-Gaussian noise models [35, 29, 34, 37, 28]. These methods first learn a topological ordering of the causal variables using the second-order derivative of the log-likelihood estimated from score matching. They then apply regression based DAG pruning techniques [10, 26] to retrieve the full causal structure. We note that these works do not inherently extend to causal disentanglement, for they assume direct access to the causal variables that disentanglement intends to learn.

Expanding these ideas to causal disentanglement is difficult, since we do not observe the latent factors and can only estimate the log-likelihood of the observed variables. Surprisingly, our theory suggests a simple principle to obtain meaningful estimates of the latent factors from the log-likelihood of the observed variables. Moreover, we show that a simple quadratic program can be used to implement this principle, which leads to an efficient algorithm borrowing strength from both nonlinear optimization and machine learning.

Our principle for identifiability directly expands the main result from [35], where variance properties on the diagonal elements of the Jacobian over the score of causal variables are used to derive a topological ordering. While we utilize this result, it is not sufficient for disentanglement given the aforementioned Jacobian can only be determined up to an unknown quadratic form when the causal variables are unobserved. Therefore, our principle additionally relies on properties of the entire Jacobian matrix, which we present in Section 3.2.

Additionally, we recognize the inherent difficulties of both non-convex optimization and second-order score estimation essential for modern score matching methods, which we discuss further in Section 4.1 and Section 5.1 respectively.

## 2 Setup

We now formally define the causal disentanglement problem and introduce relevant definitions. We consider the observed variables $X = (X_1, ..., X_d)^\top \in \mathbb{R}^{d \times 1}$ as generated from the latent causal factors $Z = (Z_1, ..., Z_n)^\top \in \mathbb{R}^{n \times 1}$ via an unknown invertible linear mixing. We do not assume that

the latent dimension $n$ is known a priori, but rather can be learned as given by the principle presented in Lemma 1 of [59]. These latent factors follow a joint distribution $p(\cdot)$, which factorizes according to an unknown directed acyclic graph (DAG) $\mathcal{G}$. We summarize the setup in the following assumption.

**Assumption 1** (**Linear mixing**). *Our data-generating process can be written as*

$$X = H \cdot Z, \qquad Z \sim p(Z) = \prod_{i=1}^{n} p(Z_i | Z_{pa(i)}),$$

*where $H \in \mathbb{R}^{d \times n}$ has full column rank and $pa(i)$ denotes the parents of node $i$ in $\mathcal{G}$.*

We assume linear mixing as it is essential for our theoretical guarantees presented in Section 3.2. However, our results also extend to settings where the true mixing function can be reduced to a linear map, such as in the case of a special class of polynomials [2, 59]. For the distribution of $Z$, we consider nonlinear additive Gaussian noise models as follows.

**Assumption 2** (**Nonlinear additive Gaussian noise model.**). *The factorization term in the joint distribution over $Z$ is specified by*

$$Z_i = f_i(Z_{pa(i)}) + \mathcal{E}_i, \qquad \forall i \in [n],$$

*where $f = \{f_i : i \in [n]\}$ are twice continuously differentiable, non-linear[2] functions that capture the dependence of $Z_i$ on its parents, and $\mathcal{E} = \{\mathcal{E}_i : i \in [n]\}$ denote exogenous noise variables, which are mutually independent and mean-zero Gaussians, i.e., $\mathcal{E}_i \sim \mathcal{N}(0, \sigma_i^2)$.*

We use $p_X(\cdot)$ to denote the induced distribution over the observed variables $X$. We consider causal disentanglement from purely observational data, where we only have access to a dataset consisting of samples from $p_X(\cdot)$. Our goal is to learn the most about $Z$ (or equivalently, $\mathcal{E}, \mathcal{G}$, and $f$) using this dataset. We additionally note that this problem has also been called *causal representation learning* in literature. Figure 1 illustrates the described setup.

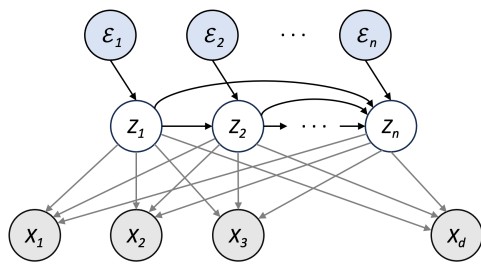

Figure 1: **The considered data-generating process.** The latent variables $Z$ follow a nonlinear causal model with additive Gaussian noises. We observe them after an unknown linear mixing (gray edges).

Figure 2: **Layers of the causal DAG.** A latent variable is contained in $layer(k)$ if its longest path to a leaf node is $k$.

**Estimators.** We denote generic estimators of $Z$ and $\mathcal{E}$ from $X$ by $\hat{Z}(X) : \mathbb{R}^d \to \mathbb{R}^n$ and $\hat{\mathcal{E}}(X) : \mathbb{R}^d \to \mathbb{R}^n$ respectively. In our setup, these estimators are constructed by learning the inverse of the unknown mixing matrix $H$. We denote a valid estimate of this mixing matrix by $\hat{H} \in \mathbb{R}^{d \times n}$, and its Moore-Penrose inverse by $\hat{H}^\dagger = \hat{H}^\top \cdot (\hat{H} \hat{H}^\top)^{-1}$. To obtain an estimate of $Z$, we use $\hat{Z}(X) = \hat{H}^\dagger \cdot X$. For simplicity, we denote the transformation from the estimated latent $\hat{Z}(X)$ to the true latent factors $Z$ by the matrix $\beta \in \mathbb{R}^{n \times n}$, where $\beta = H^\dagger \cdot \hat{H}$ and $Z = \beta \cdot \hat{Z}(X)$.

**Graph notation.** We use $ch(i)$, $an(i)$ and $de(i)$ to denote the children, ancestors and descendants of node $i$ in $\mathcal{G}$, respectively. Node $i$ is called a root node if $an(i) = \varnothing$, and a leaf node if $de(i) = \varnothing$. We define the $k^{th}$ layer of $\mathcal{G}$, denoted by $layer(k)$, to be the set of all nodes whose longest path to a leaf node is $k$. Figure 2 illustrates this concept. With a slight abuse of notation, we will interchangeably use $Z_i \in layer(k)$ to denote $i \in layer(k)$.

---

[2]To ensure that $f_i$ are non-degenerate, we assume that they are "directional non-linear", i.e., there does *not* exist $\beta \in \mathbb{R}^{|pa(i)|}$ with $\|\beta\|_2 = 1$ such that $\partial_{\beta,\beta}^2 f_i(z) = 0$ for all $z \in \mathbb{R}^{|pa(i)|}$.

# 3 Identifiability Results

In this section, we present our main theoretical results. We start by providing a precise characterization of latent factors that are identifiable in Section 3.1. We then demonstrate identifiability by providing a constructive proof in Section 3.2. Counterexamples showing that further disentanglement is not possible and our results cannot be strengthened are given in Section 3.3. Detailed proofs are deferred to Appendix A. Throughout this section, we consider the infinite-data regime where enough samples are obtained to exactly determine the observational distribution $p_X(\cdot)$.

## 3.1 Layer-wise Transformations

For each latent causal factor $Z_i$, we show that its identifiability is dependent on the layer of the corresponding node. Specifically, we show that $Z_i \in layer(k)$ can be identified up to a linear combination of all variables in $layer(k) \cup layer(k+1) \cup \cdots \cup layer(r)$, where $r$ denotes the top most layer. The formal definition is as follows.

**Definition 1** (**Identifiability up to upstream layers**). *The latent causal variables $Z$ are* identifiable up to upstream layers *if it is possible to learn $\hat{Z}(X)$ from $p_X(\cdot)$ such that:*

$$\hat{Z}(X) = P_\pi \cdot C \cdot Z, \qquad \forall Z \in \mathbb{R}^n,$$

*where $P_\pi \in \mathbb{R}^{n \times n}$ is a permutation matrix, and $C \in \mathbb{R}^{n \times n}$ is a constant matrix with non-zero diagonal entries and $[C]_{i,j} = 0$ for all $i, j$ such that $i \in layer(k)$ and $j \in \cup_{l \leq k} layer(l)$.*

This identifiability notion implies that each causal variable can be learned up to a linear combination that does not depend on its descendants. Intuitively, this implies that variables that are more upstream in the underlying causal DAG can more easily be identified. In particular, the root nodes (i.e., the most upstream causal factors) can be identified up to a linear transformation of themselves.

Beyond this $Z$-based notion of identifiability, we can further disentangle the exogenous noise variables up to a transformation that depends only on its own layer. The formal definition is as follows.

**Definition 2** (**Identifiability up to layers**). *The exogenous noise variables $\mathcal{E}$ are* identifiable up to layers *if it is possible to learn $\hat{\mathcal{E}}(X)$ from $p_X(\cdot)$ such that:*

$$\hat{\mathcal{E}}(X) = P_\pi \cdot C \cdot \mathcal{E}, \qquad \forall \mathcal{E} \in \mathbb{R}^n,$$

*where $P_\pi \in \mathbb{R}^{n \times n}$ is a permutation matrix, and $C \in \mathbb{R}^{n \times n}$ is a constant matrix with non-zero diagonal entries and $[C]_{i,j} = 0$ for all $i, j$ such that $i \in layer(k)$ and $j \notin layer(k)$.*

Next, we prove these notions of identifiability in a constructive way using the score function of $p_X(\cdot)$.

## 3.2 Identification via Score Functions

Our analysis will rely on the *score function* of the observational distribution of $X$, denoted by

$$s_X(x) = \nabla_x \log p_X(x),$$

as well as its *Jacobian matrix* whose $ij^{th}$ entry is given by $[J_X(x)]_{ij} = \nabla_{x_i} \nabla_{x_j} \log p_X(x)$. Since $X$ and $Z$ are related through a linear transformation, we can easily write out the closed form for both the score and associated Jacobian of the latent variables $Z$ as follows.

**Lemma 1.** [3] *Under Assumption 1, the score functions and associated Jacobian matrices over $X$ and $Z$ are related via the following transformations:*

$$s_Z(z) = H^\top s_X(x), \qquad J_Z(z) = H^\top J_X(x) H.$$

For an estimator $\hat{Z}(X) = \hat{H} \cdot X$, we utilize Lemma 1 to obtain the following:

$$J_{\hat{Z}}(\hat{z}) = \hat{H}^\top J_X(x) \hat{H} = \beta^\top J_Z(z) \beta.$$

---

[3]Similar results have been used in [48, 49], where [49] provided formulas for general mixings.

This shows that we can compute $J_{\hat{Z}}$ once we estimate $\hat{H}$ and $J_X$ from $p_X(\cdot)$, and that $J_{\hat{Z}}$ relates to the Jacobian matrix over the true latent variables, $J_Z$, via a quadratic form $J_{\hat{Z}} = \beta^\top J_Z \beta$, where $\beta$ is a product of the unknown $H^\dagger$ and $\hat{H}$.

Under the nonlinear additive Gaussian noise model in Assumption 2, [35] demonstrated that the $i^{th}$ diagonal element of $J_Z$ will have zero variance if and only if node $i$ is a leaf node in $\mathcal{G}$. Building on this result, we can derive a sufficient and necessary condition for when the $i^{th}$ diagonal element of $J_{\hat{Z}}$ will have zero variance involving the unknown matrix $\beta$ as follows.

**Lemma 2.** *The $i^{th}$ diagonal element of $[J_{\hat{Z}}(\hat{z})]_{ii}$ has zero variance, i.e., $\mathrm{Var}\left([J_{\hat{Z}}(\hat{z})]_{ii}\right) = 0$, if and only if the $i^{th}$ column of $\beta$ has zero entries in every element corresponding to non-leaf nodes.*

This result provides the intuition that leads to the principle for achieving identifiability. In particular, if we maximize the number of zero-variance terms in the diagonal elements of the estimated Jacobian $J_{\hat{Z}}$, then the unknown matrix $\beta$ must have a maximum number of columns with zeros in all indices corresponding to non-leaf nodes. Since $Z = \beta \cdot \hat{Z}$, we can derive the relation between $\hat{Z}$ and $Z$ under this maximization, which we summarize in the following lemma.

**Lemma 3.** *If we learn $\hat{H}$ by solving*

$$\min_{\hat{H} \in \mathbb{R}^n} \quad \left\|\mathrm{Var}\left(\mathrm{diag}(J_{\hat{Z}}(\hat{H}^\dagger x))\right)\right\|_0,$$

$$\text{such that} \quad \mathrm{rank}(\hat{H}) = n, \tag{1}$$

*then it follows that*

$$\hat{Z}_i = \begin{cases} \mathrm{linear}(Z_{non-leaf}) & \textit{if} \quad \mathrm{Var}\left([J_{\hat{Z}}(\hat{z})]_{ii}\right) \neq 0, \\ \mathrm{linear}(Z) & \textit{if} \quad \mathrm{Var}\left([J_{\hat{Z}}(\hat{z})]_{ii}\right) = 0, \end{cases}$$

*where the number of $i \in [n]$ such that $\mathrm{Var}\left([J_{\hat{Z}}(\hat{z})]_{ii}\right) = 0$ equals to the number of leaf nodes in $\mathcal{G}$.*

It follows from this lemma that we can obtain representations of all non-leaf nodes as linear transformations of all non-leaf latent variables (i.e. $layer(1)$ and above). In other words, we can disentangle the leaf nodes out from the non-leaf nodes. Given this identified linear transformation of the non-leaf nodes, we can iteratively apply Lemma 3 to prune representations of each variable as a linear combination of all variables in its own and upstream layers. This leads to our main theorem.

**Theorem 1.** *Under Assumptions 1 and 2, the latent variables $Z$ are identifiable up to their upstream layers from purely observational data.*

Importantly, this result holds without any structural restrictions on the mixing function or the latent causal DAG. It indicates that we can derive representations of latent factors free of all downstream variables, and that it is easier to disentangle the more upstream causal factors.

Building on Theorem 1, we can show a stronger notion of identifiability for the exogenous noise variables. Consider any $layer(i)$ representation given by a linear combination of all variables in $layer(i+1) \cup \cdots \cup layer(r)$, where $r$ denotes the top most layer. Then from the structural equations, it follows that this representation depends nonlinearly on the exogenous noise variables associated with $layer(i+1) \cup \cdots \cup layer(r)$ and linearly on the the exogenous noise variables associated with $layer(i)$, which we denote by $\mathcal{E}_{layer(i)}$. Thus, if we regress this representation on all upstream layer representations, e.g., using kernel regression, then the residual terms will equate to a linear combination of $\mathcal{E}_{layer(i)}$. Performing this procedure over all layers $i$, we can determine layer-wise transformations of $\mathcal{E}$, giving rise to the following theorem.

**Theorem 2.** *Under Assumptions 1 and 2, the exogenous noise variables $\mathcal{E}$ are identifiable up to their layers from purely observational data.*

### 3.3 Impossibility Results

Next, we show that further disentanglement is not possible. In particular, we cannot further disentangle the exogenous noise variables within any given layer. The following example illustrates this with two variables. Suppose the exogenous noise variables $\mathcal{E}_1$ and $\mathcal{E}_2$ are identified via two linear combinations denoted by

$$\hat{\mathcal{E}}_1 = a_1 \mathcal{E}_1 + a_2 \mathcal{E}_2, \quad \hat{\mathcal{E}}_2 = b_1 \mathcal{E}_1 + b_2 \mathcal{E}_2.$$

We only know that they are independent mean-zero Gaussian variables. However, any linear coefficients with $a_1 b_1 \sigma_1^2 + a_2 b_2 \sigma_2^2 = 0$ satisfy $Cov(\hat{\mathcal{E}}_1, \hat{\mathcal{E}}_2) = a_1 b_1 \sigma_1^2 + a_2 b_2 \sigma_2^2 = 0$, which means $\hat{\mathcal{E}}_1$ and $\hat{\mathcal{E}}_2$ are independent. This indicates that $\hat{\mathcal{E}}_1$ and $\hat{\mathcal{E}}_2$ do not provide enough information to further disentangle $\mathcal{E}_1$ or $\mathcal{E}_2$. In general, this impossibility result holds for arbitrary graphs.

**Proposition 1.** *Under Assumptions 1 and 2, the exogenous noise variables $\mathcal{E}$ are generally unidentifiable beyond layer-wise transformation from observational data.*

## 4 Algorithm for Layer Recovery

We now transition to developing practical algorithms to recover the guaranteed causal representations. Our approach consist of two steps: (1) solving for the representations of latent variables up to upstream-layer transformations, and (2) solving for representations of exogenous noise variables up to layer-wise transformations, where step 2 utilizes the output of step 1.

### 4.1 Step 1: Quadratic Programming on Estimated Scores

The proof sketch in Section 3.2 provides a simple principle for causal disentanglement. It suggests that we can solve for the estimated mixing function, $\hat{H}$, at each iteration by maximizing the number of zero-variance terms in the Jacobian of the estimated latent score (i.e., Equation (1)). However, this rank-constrained optimization problem is discontinuous and non-convex, leading to an NP-hard problem [46]. Moreover, the objective function involves the $\ell_0$-norm of a vector of variance terms, which can be hard to optimize.

To resolve these difficulties, we reduce this optimization problem into a sequence of easier problems. Note that $Var[J_{\hat{Z}}(z)]_{ii}$ depends only on the $i^{th}$ column of $\hat{H}$, which we denote as $[\hat{H}]_i$. It follows that we can solve for each column separately by solving for $[\hat{H}]_i$ such that $Var[J_{\hat{Z}}(z)]_{ii} = 0$ while not violating the rank constraint. Considering the finite-sample setting where we plug in the sample estimate for $Var[J_{\hat{Z}}(z)]_{ii}$, this problem can be formulated as the following quadratically constrained quadratic program (QCQP):

$$
\begin{aligned}
\min_{h \in \mathbb{R}^n} \quad & 0 \\
\text{such that} \quad & h^\top \tilde{J}_X(x^{(m)}) h = 0, \quad \forall m \in [N], \\
& h^\top h = 1, \\
& h^\top [\hat{H}]_j = 0, \quad \forall j \in [i-1].
\end{aligned}
\tag{2}
$$

Here, we use estimated zero-centered Jacobians $\tilde{J}_X$ of observed samples $x^{(m)}$, given by $\tilde{J}_X(x^{(m)}) \triangleq \hat{J}_X(x^{(m)}) - \bar{J}_X(X)$ with $\bar{J}_X(X) \triangleq 1/N \sum_{m=1}^N \hat{J}_X(x^{(m)})$. The constraint $h^\top \tilde{J}_X(x^{(m)}) h = 0$ is equivalent to enforcing the sample estimate of $Var[J_{\hat{Z}}(z)]_{ii}$ to be zero. The additional constraints $h^\top h = 1$ and $h^\top [\hat{H}]_j = 0$ ensure that we do not violate the rank constraint. A formal derivation of equivalence is given in Appendix B.

Breaking the problem in Equation (1) into a series of problems in Equation (2) allows us to operate over a lower dimensional space and use any off-the-shelf solvers for QCQP. In practice, we use the cutting plane method for mixed integer programming [25]. Algorithm 1 summarizes the overall approach, where we construct the $layer(k)$ representations iteratively by solving a series of QCQPs.

### 4.2 Step 2: Layer-wise Nonlinear Regression

Given the learned representation $\hat{Z}$ from Algorithm 1, we now proceed to disentangle the exogenous noise variables, which fully determine the randomness of the observations.

Following the proof of Theorem 2, we can recover a representation of the exogenous noise variables $\mathcal{E}_{layer(k)}$ by non-linearly regressing the $layer(k)$ representation of $Z$ on all upstream representations and taking the residual terms. This procedure is summarized in Algorithm 2.

---

**Algorithm 1** Recovering $Z$ up to upstream layers.

---

1: **Input:** $N$ samples of $X$ in the observational distribution.
2: Estimate $\tilde{J}_X(x^{(m)}), \forall m \in [N]$ using any off-the-shelf score estimation method (see Section 5).
3: Initialize $\hat{Z} = 0^{n \times N}$, $\hat{X} = (x^{(1)}, \ldots, x^{(N)}) \in \mathbb{R}^{d \times N}$, and $k = n$.
4: **while** $k > 0$ **do**
5:    Initialize $\hat{H} = 0^{k \times d}$.
6:    **for** $i = 1, \ldots, d$ **do**
7:       Set $[\hat{H}]_i$ to be the solution of Equation (2). Break when no feasible solution is found.
8:    **end for**
9:    Fill in all-zero columns of $\hat{H}$ with random vectors to remain full column rank.
10:    Compute $\tilde{Z} = \hat{H}^\dagger \hat{X}$ and $J_{\tilde{Z}}(\tilde{Z}^{(m)}) = \hat{H}^\top \tilde{J}_X(\hat{x}^{(m)}) \hat{H}, \forall m \in [N]$.
11:    Set $\hat{X} = 0^{0 \times N}$.
12:    **for** $i = 1, ..., k$ **do**
13:       **if** $Var[J_{\hat{Z}}(\hat{z})]_{i,i} = 0$ **then**
14:          Set $[\hat{Z}]_{n-k} = [\tilde{Z}]_i$ and let $k \leftarrow k - 1$.
15:       **else**
16:          $\hat{X} \leftarrow \left[\hat{X}, [\tilde{Z}]_i\right]$.
17:       **end if**
18:    **end for**
19: **end while**
20: **Return:** $\hat{Z}$

---

**Algorithm 2** Recovering $\mathcal{E}$ up to layers.

---

1: **Input:** $\hat{Z} \in \mathbb{R}^{n \times N}$ estimated from Algorithm 1. Denote the number of layers as $K$.
2: Initialize $\hat{\mathcal{E}} = 0^{n \times N}$. Set $\hat{\mathcal{E}}_{layer(K)} = \hat{Z}_{layer(K)}$.
3: **for** $k = K - 1, ..., 0$ **do**
4:    Fit nonlinear regression on $\hat{Z}_{layer(k)}$ using $\hat{\mathcal{E}}_{layer(k+1)}, \ldots, \hat{\mathcal{E}}_{layer(K)}$.
5:    Set $\hat{\mathcal{E}}_{layer(k)}$ as the residual terms.
6: **end for**
7: **Return:** $\hat{\mathcal{E}}$.

---

# 5    Numerical Results

We test our proposed algorithms using simulations[4]. Algorithm 1 requires estimating the score of the observational distribution and its performance relies on the quality of this estimation. To evaluate this, we conduct two sets of experiments. In Section 5.1, we use perfect score oracles, which compute the Jacobian matrices exactly using the ground-truth data-generating process. This serves as a verification of our theoretical results. In Section 5.2, we estimate the score functions from the samples using two popular score-estimation methods. Details of the experiments can be found in Appendix C.

## 5.1    Score Oracle Simulations: Validation of Theoretical Results

To further validate our theoretical results, we run Algorithms 1 and 2 to learn the latent causal factors and the exogenous noise variables. We consider the following causal graphs with 4 nodes: (1) a line graph represented as $Z_1 \to Z_2 \to Z_3 \to Z_4$, and (2) a Y-structure represented as $Z_1 \to Z_2 \to Z_3$, $Z_2 \to Z_4$. For each case, we generate 2000 observational samples and compute the corresponding scores using the ground-truth link functions.

We present the results of our estimation in Figure 3. The scatter plots depict the relationships between the ground-truth $\mathcal{Z}_i$ and the estimated $\hat{\mathcal{Z}}_j$, where we color the dots with the values of $Z_1$. The heatmaps show the mean absolute correlations (MAC) between the ground-truth $\mathcal{E}_i$ and the estimated $\hat{\mathcal{E}}_j$. For the estimated latent causal factors $\hat{Z}$, we see trends that are consistent with Theorem 1 in both cases, where the root node is perfectly identified and $Z_2$ is estimated with some mixing of $Z_1$.

---

[4]Code is publicly available at: `https://github.com/uhlerlab/observational-crl`

For the estimated exogenous noise variables $\hat{\mathcal{E}}$, the results validate Theorem 2. In the line graph, our algorithm perfectly disentangles all variables; in the Y-structure, we can perfectly disentangle $\mathcal{E}_1$ and $\mathcal{E}_2$, while $\mathcal{E}_3$ and $\mathcal{E}_4$ are mixed.

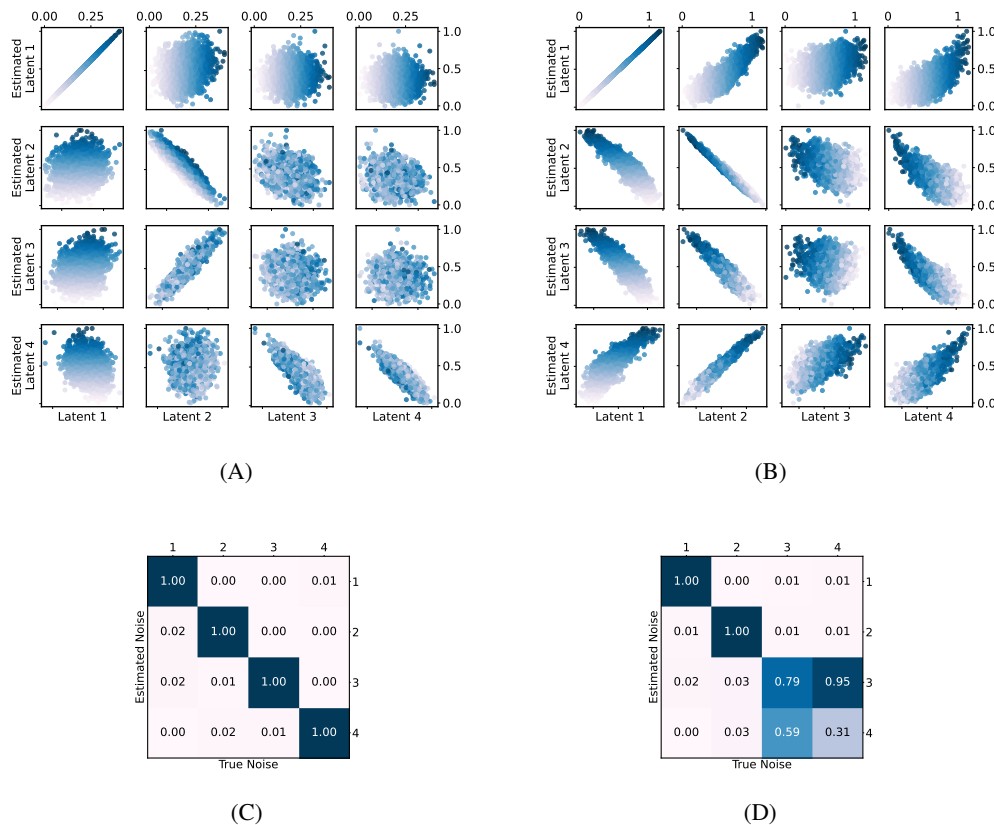

Figure 3: **Score oracle simulations.** (A) Estimated versus true latent variables on the line graph. (B) Estimated versus true latent variables on the Y-structure. (C) Estimated versus true exogenous variables on the line graph. (D) Estimated versus true exogenous variables on the Y-structure.

## 5.2 Results using Score Estimation

In this set of experiments, we aim to mimic real-world settings where the score functions are estimated from samples. We use two popular methods to generate point-wise estimates of the Jacobians of the scores: the second-order Stein estimator [5, 44] and the sliced score matching with variance reduction (SSM-VR) estimator [42]. We then plug these estimators into Algorithms 1 and 2.

We use the same sampling procedure on the four-node line graph as described in the previous section with varying sample sizes. Here we evaluate the mean absolute correlation (MAC) between the true and estimated exogenous noise variables. We adjust the tolerance of our QCQP solver to account for noisy estimates (see Appendix C). Table 2 reports the results averaged across 10 repeated runs. With noisy score estimates, we can still learn these variables although, as expected, accuracy decreases as compared to the results using oracle score estimates, where we can recover the exogenous noise almost perfectly.

As reliable higher-order score estimation is an active area of research [27, 41, 30], we seek to evaluate how the accuracy of our algorithm can increase under improved score estimation. Specifically, we consider how the MAC of exogenous noise estimates behaves under varying levels of noise in the plug-in Jacobians. We perturb the true Jacobian matrices with noise and plot the returned MAC with respect to the signal-to-error ratio (SER) in Figure 4. This shows that the accuracy improves with

Table 2: Mean absolute correlation of the exogenous noise estimates using score estimations.

| Samples | Oracle | Stein | SSM-VR |
|---------|--------|-------|--------|
| $1 \times 10^3$ | $0.999 \pm 0.0$ | $0.39 \pm 0.093$ | $0.358 \pm 0.153$ |
| $5 \times 10^3$ | $1.0 \pm 0.0$ | $0.445 \pm 0.164$ | $0.45 \pm 0.143$ |
| $1 \times 10^4$ | $1.0 \pm 0.0$ | $0.371 \pm 0.227$ | $0.44 \pm 0.222$ |
| $5 \times 10^4$ | $1.0 \pm 0.0$ | $0.367 \pm 0.135$ | $0.43 \pm 0.115$ |

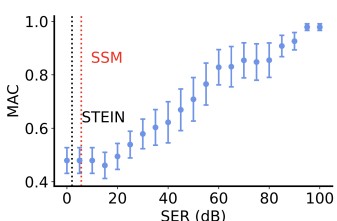

Figure 4: Mean absolute correlation (MAC) of $\mathcal{E}$ estimations v.s. Signal-to-error ratio (SER) of Jacobian matrices.

higher SER. We also mark the MAC of Stein estimation and SSM-VR, which are approximately around 2 and 6 SERs respectively.

## 6   Discussion

In this work, we derive partial identifabilty guarantees of causal disentanglement from purely observational data and linear mixing without any structural restrictions. In particular, we utilize asymmetries in nonlinear causal models with additive Gaussian noise. We provide a precise characterization of identifiability in this setting, where the latent causal factors can be identified up to upstream layers and the exogenous noise variables can be identified up to their layers. We show that further disentanglement is not possible without additional assumptions or alternative datasets.

These theoretical analyses indicate a simple but hard to optimize principle for deriving efficient algorithms. We show that this optimization problem can be solved via a series of simpler quadratically constrained quadratic programs. This leads to a flexible algorithm that allows us to use any off-the-shelf QCQP solvers and score estimation methods. We demonstrate its correctness and efficiency using simulations.

While we view this work as having a primarily theoretical contribution, we additionally believe that the notion of layer-wise identifiability has many practical implications. In particular, our methods can be used to identify hierarchical topics at various layers of a causal system. For instance, when applied to a latent genealogical tree, each layer representation would contain all prior ancestral information used to determine the traits of a given generation.

In future work, it would be interesting to extend our results to latent causal models with other asymmetries. In particular, we believe our result could be extended to learn upstream layer representations of nonlinear additive models with generic noise as an extension of [28], by modifying the principle to achieve identifiability in Equation (1). It would also be interesting to understand how our identifability results in the purely observational setting could aid when additional external data, such as interventions or multi-modal data, are available. Additionally, since our work shows that causal disentanglement can be solved orthogonally to score estimation, extending and testing our proposed approaches to applications where there exist pretrained score estimators would be another interesting avenue to pursue. Furthermore, given further disentanglement beyond layer-wise identifiability is not possible with purely observational data, it remains unclear what minimum faithfulness assumptions are required to achieve stronger identifiability guarantees, which we view as an import question.

**Broader impact.** Our work advances the field of causal representation learning, where it was commonly thought that without interventional data causal variables could only be discovered up to linear combinations of all variables without structural assumptions. While our work has many potential applications, we feel that no particular societal consequence needs to be highlighted.

## Acknowledgements

We thank Chandler Squires for helpful discussions, as well as Karthikeyan Shanmugan and the anonymous reviewers for their valuable feedback. This work was partially supported by ONR (N00014-22-1-2116), NCCIH/NIH (1DP2AT012345), DOE (DE-SC0023187), and a Simons Investi-

gator Award to Caroline Uhler. R.W. was supported by a fellowship by the Eric and Wendy Schmidt Center at the Broad Institute and the Advanced Undergraduate Research Opportunities Program at MIT. J.Z. was partially supported by an Apple AI/ML PhD Fellowship.

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

# A  Proofs for Identifiability

## A.1  Proof of Lemma 1

*Proof.* Given the linear relation $X = H \cdot Z$, we relate the probability density functions $p(\cdot)$ of $Z$ and $p_X(\cdot)$ via

$$p_X(x) = p(H^\dagger x)|\det(H^\dagger)|.$$

Furthermore, we write the gradient of the log density of $p_X(x)$ with respect to $X$ as

$$
\begin{aligned}
\nabla_X \log p_X(x) &= \frac{\nabla_X p_X(x)}{p_X(x)} \\
&= \frac{\nabla_X p(H^\dagger x)|\det(H^\dagger)|}{p(H^\dagger x)|\det(H^\dagger)|} \\
&= \frac{\nabla_X p(H^\dagger x)}{p(H^\dagger x)} \\
&= \nabla_X \log p(H^\dagger x) \\
&= (H^\dagger)^\top \nabla_Z \log p(z).
\end{aligned}
$$

Thus, it follows that $\nabla_Z \log p(z) = H^\top \nabla_X \log p_X(x)$, or $s_Z(z) = H^\top s_X(x)$ as desired.

Differentiating $\nabla_X \log p_X(x)$ with respect to $X$, we get

$$
\begin{aligned}
\nabla_X^2 \log p_X(x) &= \nabla_X (H^\dagger)^\top \nabla_Z \log p(z) \\
&= (H^\dagger)^\top \nabla_Z^2 \log p(z)(H^\dagger).
\end{aligned}
$$

Thus, it additionally follows that $\nabla_Z^2 \log p(z) = H^\top (\nabla_X^2 \log p_X(x))H$, or $J_Z(z) = H^\top J_X(x)H$.
$\square$

## A.2  Proof of Lemma 2

Before proceeding to the proof of Lemma 2, we must prove the following supplementary lemma.

**Lemma 4.** *For any two distinct leaf nodes $k$ and $l$, it follows that $\frac{\partial s_k(z)}{\partial z_l} = 0$.*

*Proof.* From [35], we denote the score of the latent variable $Z_k$ evaluated at $z$ as

$$s_k(z) = -\frac{z_k - f_k(z_{pa(k)})}{\sigma_k^2} + \sum_{i \in ch(k)} \frac{\partial f_i(z_{pa(i)})}{\partial z_k} \cdot \frac{z_i - f_i(z_{pa(i)})}{\sigma_i^2}.$$

We further derive the following expression:

$$
\begin{aligned}
&\frac{\partial s_k(z)}{\partial z_l} \\
&= \left(\frac{1}{\sigma_k^2}\right)\left(\frac{\partial f_k(z_{pa(k)})}{\partial z_l}\right) \\
&\quad + \sum_{i \in ch(k)}\left[\left(\frac{\partial^2 f_i(z_{pa(i)})}{\partial z_k \partial z_l}\right)\left(\frac{z_i - f_i(z_{pa(i)})}{\sigma_i^2}\right) + \left(\frac{1}{\sigma_i^2}\right)\left(\frac{\partial f_i(z_{pa(i)})}{\partial z_k}\right)\left(\frac{\partial z_i}{\partial z_l} - \frac{\partial f_i(z_{pa(i)})}{\partial z_l}\right)\right] \\
&= \left(\frac{1}{\sigma_k^2}\right)\left(\frac{\partial f_k(z_{pa(k)})}{\partial z_l}\right) \\
&\quad + \sum_{i \in ch(k)}\left[\left(\frac{\partial^2 f_i(z_{pa(i)})}{\partial z_k \partial z_l}\right)\left(\frac{z_i - f_i(z_{pa(i)})}{\sigma_i^2}\right) + \left(\frac{1}{\sigma_i^2}\right)\left(\frac{\partial f_i(z_{pa(i)})}{\partial z_k}\right)\left(\mathbf{1}_{\{l=i\}} - \frac{\partial f_i(z_{pa(i)})}{\partial z_l}\right)\right] \\
&= \left(\frac{1}{\sigma_k^2}\right)\left(\frac{\partial f_k(z_{pa(k)})}{\partial z_l}\right) + \mathbf{1}_{\{l \in ch(k)\}}\left(\frac{1}{\sigma_l^2}\right)\left(\frac{\partial f_l(z_{pa(l)})}{\partial z_k}\right) + \\
&\quad \sum_{i \in ch(k) \cap ch(l)} \frac{1}{\sigma_i^2}[\nabla_k \nabla_l f_i(z_{pa(i)}) \cdot z_i - \nabla_k \nabla_l f_i(z_{pa(i)}) \cdot f_i(z_{pa(i)}) - \nabla_k f_i(z_{pa(i)}) \cdot \nabla_l f_i(z_{pa(i)})].
\end{aligned}
$$

When $k$ and $l$ are distinct leaf nodes, $ch(k) = ch(l) = \varnothing$, and our expression simplifies to

$$\frac{\partial s_k(z)}{\partial z_l} = \left(\frac{1}{\sigma_k^2}\right)\left(\frac{\partial f_k(z_{pa(k)})}{\partial z_l}\right) = 0,$$

since $l \notin pa(k)$, which completes our proof. $\qquad\square$

We now proceed to the proof of Lemma 2.

*Proof.* (Lemma 2) We first prove the backward direction. Given $J_{\hat{Z}}(\hat{z}) = \beta^\top J_Z(z)\beta$, we express $J_{\hat{Z}}(\hat{z})_{ii}$ as

$$J_{\hat{Z}}(\hat{z})_{ii} = \sum_{j=1}^n \sum_{k=1}^n \beta_{ji}\beta_{ki}\frac{\partial s_j(z)}{\partial z_k}.$$

Assuming that $\beta_{ji} = 0$ for all $j \notin layer(0)$, the above expression simplifies to

$$J_{\hat{Z}}(\hat{z})_{ii} = \sum_{j,k \in layer(0)} \beta_{ji}\beta_{ki}\frac{\partial s_j(z)}{\partial z_k}.$$

Utilizing Lemma 4 and Lemma 1 from [35], which states that $\text{Var}\left[\frac{\partial s_k(z)}{\partial z_k}\right] = 0$ for all $k \in layer(0)$, it follows that $\text{Var}\left[J_{\hat{Z}}(\hat{z})_{ii}\right] = 0$.

Now, we prove the forward direction. Denote $C_i := \{j : \beta_{ji} \neq 0\}$ as the set of all indices in the $i^{th}$ column of $\beta$ that are non-zero. It suffices to show that if there exists some $k \in C_i$ such that $k \notin layer(0)$, then $\text{Var}\left[J_{\hat{Z}}(\hat{z})_{ii}\right] \neq 0$.

Let $i_c$ be the most downstream node in $ch(C_i) := \{ch(j) : j \in C_i\}$ such that $i_c \notin pa(i)$ for any $i \in ch(C_i)$. Such a node must exist under the assumption that some non-leaf node is contained in $C_i$. We express $J_{\hat{Z}}(\hat{z})_{ii}$ as follows

$$J_{\hat{Z}}(\hat{z})_{ii}$$

$$= \sum_{k,l \in C_i} \beta_{ki}\beta_{li}\frac{\partial s_k(z)}{\partial z_l} = \sum_{k \in C_i} \beta_{ki}^2 \frac{\partial s_k(z)}{\partial z_k} + \sum_{\substack{k,l \in C_i \\ k \neq l}} \beta_{ki}\beta_{li}\frac{\partial s_k(z)}{\partial z_l}$$

$$= \sum_{k \in C_i} \beta_{ki}^2 \left[-\frac{1}{\sigma_k^2} + \sum_{i \in ch(k)} \frac{1}{\sigma_i^2}\left[\nabla_k^2 f_i(z_{pa(i)}) \cdot z_i - \nabla_k^2 f_i(z_{pa(i)}) \cdot f_i(z_{pa(i)}) - (\nabla_k f_i(z_{pa(i)}))^2\right]\right]$$

$$+ \sum_{\substack{k,l \in C_i \\ k \neq l}} \beta_{ki}\beta_{li}\left[\left(\frac{1}{\sigma_k^2}\right)\left(\frac{\partial f_k(z_{pa(k)})}{\partial z_l}\right) + 1_{\{l \in ch(k)\}}\left(\frac{1}{\sigma_l^2}\right)\left(\frac{\partial f_l(z_{pa(l)})}{\partial z_k}\right) + \right.$$

$$\left. \sum_{i \in ch(k) \cap ch(l)} \left[\frac{1}{\sigma_i^2}\nabla_k \nabla_l f_i(z_{pa(i)}) \cdot z_i - \nabla_k \nabla_l f_i(z_{pa(i)}) \cdot f_i(z_{pa(i)}) - \nabla_k f_i(z_{pa(i)}) \cdot \nabla_l f_i(z_{pa(i)})\right]\right]$$

$$= \sum_{k \in C_i} \beta_{ki}^2 \left[-\frac{1}{\sigma_k^2} + \sum_{i \in ch(k)} \frac{1}{\sigma_i^2}\left[\nabla_k^2 f_i(z_{pa(i)}) \cdot \mathcal{E}_i - (\nabla_k f_i(z_{pa(i)}))^2\right]\right] +$$

$$\sum_{\substack{k,l \in C_i \\ k \neq l}} \beta_{ki}\beta_{li}\left[\left(\frac{1}{\sigma_k^2}\right)\left(\frac{\partial f_k(z_{pa(k)})}{\partial z_l}\right) + 1_{\{l \in ch(k)\}}\left(\frac{1}{\sigma_l^2}\right)\left(\frac{\partial f_l(z_{pa(l)})}{\partial z_k}\right) + \right.$$

$$\left. \sum_{i \in ch(k) \cap ch(l)} \left[\frac{1}{\sigma_i^2}\nabla_k \nabla_l f_i(z_{pa(i)}) \cdot \mathcal{E}_i - \nabla_k f_i(z_{pa(i)}) \cdot \nabla_l f_i(z_{pa(i)})\right]\right].$$

Now, by separating the terms containing $i_c$, we get

$$J_{\hat{Z}}(\hat{z})_{ii} = \sum_{\substack{k \in C_i \\ k \in pa(i_c)}} (\beta_{ki}^2) \left(\frac{1}{\sigma_{i_c}^2}\right) (\nabla_k^2 f_{i_c}(z_{pa(i_c)})) (\mathcal{E}_{i_c}) + \tag{3}$$

$$\sum_{\substack{k,l \in C_i \\ k \neq l \\ k,l \in pa(i_c)}} (\beta_{ki}\beta_{li}) \left(\frac{1}{\sigma_{i_c}^2}\right) (\nabla_k \nabla_l f_{i_c}(z_{pa(i_c)})) (\mathcal{E}_{i_c}) - \tag{4}$$

$$\sum_{\substack{k \in C_i \\ k \in pa(i_c)}} (\beta_{ki}^2) \left(\frac{1}{\sigma_{i_c}^2}\right) (\nabla_k f_{i_c}(z_{pa(i_c)}))^2 - \tag{5}$$

$$\sum_{\substack{k,l \in C_i \\ k \neq l \\ k,l \in pa(i_c)}} (\beta_{ki}\beta_{li}) \left(\frac{1}{\sigma_{i_c}^2}\right) (\nabla_k f_{i_c}(z_{pa(i_c)})) (\nabla_l \nabla_l f_{i_c}(z_{pa(i_c)})) + \tag{6}$$

$$\sum_{k \in C_i} \beta_{ki}^2 \left[ -\frac{1}{\sigma_k^2} + \sum_{\substack{i \in ch(k) \\ i \neq i_c}} \frac{1}{\sigma_i^2} \left[ \nabla_k^2 f_i(z_{pa(i)}) \cdot \mathcal{E}_i - (\nabla_k f_i(z_{pa(i)}))^2) \right] \right] + \tag{7}$$

$$\sum_{\substack{k \in C_i \\ k \neq l}} \beta_{ki}\beta_{li} \left[ \left(\frac{1}{\sigma_k^2}\right) (\nabla_l f_k(z_{pa(k)})) + 1_{\{l \in ch(k)\}} \left(\frac{1}{\sigma_l^2}\right) (\nabla_k f_l(z_{pa(l)})) + \tag{8}$$

$$\sum_{\substack{i \in ch(k) \cap ch(l) \\ i \neq i_c}} \left[ \frac{1}{\sigma_i^2} \nabla_k \nabla_l f_i(z_{pa(i)}) \cdot \mathcal{E}_i - \nabla_k f_i(z_{pa(i)}) \cdot \nabla_l f_i(z_{pa(i)}) \right] \right]. \tag{9}$$

Let $g(z_{-i_c}) = (5) + (6) + (7) + (8) + (9)$. Given that (5), (6), (7), (8) and (9) are functions of only variables upstream of $Z_{i_c}$, we have that $g(z_{-i_c}) \perp\!\!\!\perp \mathcal{E}_{i_c}$. Furthermore, let

$$h(z_{-i_c}) = \sum_{\substack{k,l \in C_i \\ k,l \in pa(i_c)}} (\beta_{ki}\beta_{li}) \left(\frac{1}{\sigma_{i_c}^2}\right) (\nabla_k \nabla_l f_{i_c}(z_{pa(i_c)})),$$

which similarly contains only variables upstream of $z_{i_c}$. Thus it holds that $h(z_{-i_c}) \perp\!\!\!\perp \mathcal{E}_{i_c}$. Now we write

$$J_{\hat{Z}}(\hat{z})_{ii} = h(z_{-i_c}) \cdot \mathcal{E}_{i_c} + g(z_{-i_c}),$$

and it suffices to show that $Var[h(z_{-i_c}) \cdot \mathcal{E}_{i_c} + g(z_{-i_c})] \neq 0$. Expanding this expression, we get

$$Var[h(z_{-i_c}) \cdot \mathcal{E}_{i_c} + g(z_{-i_c})] = Var[h(z_{-i_c}) \cdot \mathcal{E}_{i_c}] + Var[g(z_{-i_c})] + 2Cov[h(z_{-i_c}) \cdot \mathcal{E}_{i_c}, g(z_{-i_c})]$$
$$= \mathbb{E}[h(z_{-i_c})^2 \mathcal{E}_{i_c}^2] - \mathbb{E}[[h(z_{-i_c})\mathcal{E}_{i_c}]^2 + Var(g) + 2\mathbb{E}[h(z_{-i_c})\mathcal{E}_{i_c} g(z_{-i_c})]$$
$$- 2\mathbb{E}[h(z_{-i_c})\mathcal{E}_{i_c}]\mathbb{E}[g(z_{-i_c})]$$
$$= \mathbb{E}[h(z_{-i_c})^2]\mathbb{E}[\mathcal{E}_{i_c}^2] + Var(g)$$
$$= Var[h(z_{-i_c})]\sigma_{i_c}^2 + \mathbb{E}[h(z_{-i_c})]^2 \sigma_{i_c}^2 + Var(g).$$

Therefore, the variance of $J_{\hat{Z}}(\hat{z})_{ii}$ is positive if and only if $Var[h(z_{-i_c})] > 0$, $\mathbb{E}[h(z_{-i_c})] \neq 0$, or $Var(g(z_{-i_c})) > 0$. We will now show that $Var[h(z_{-i_c})] > 0$ or $\mathbb{E}[h(z_{-i_c})] \neq 0$.

Let us rewrite $h(z_{-i_c})$ in matrix form as $h(z_{-i_c}) \propto (\beta_i')^\top \partial^2_{z_{pa(i_c)}, z_{pa(i_c)}} f_{i_c}(z_{pa(i_c)})(\beta_i') = \partial^2_{\beta_i', \beta_i'} f_{i_c}(z_{pa(i_c)})$, where $\beta_i' \in \mathbb{R}^{|pa(i_c)|}$ is a vector formed by finding the entries that correspond to $pa(i_c)$ in $\beta$. Note that by our assumption $pa(i_c) \cap C_i \neq \varnothing$, we thus have $\beta_i' \neq 0$. By the directional non-linear assumption in Assumption 2, we know that $\partial^2_{\beta_i', \beta_i'} f_{i_c}(z_{pa(i_c)})$ cannot be 0 for all realizations of $z_{pa(i_c)}$. This implies that $\mathbb{P}[h(z_{-i_c}) = 0] \neq 1$, which further implies that either $Var[h(z_{-i_c})] > 0$ or $\mathbb{E}[h(z_{-i_c})] \neq 0$ as desired. $\qquad \square$

## A.3 Proof of Lemma 3

*Proof.* Without loss of generality, assume that $layer(0)$ corresponds to the the last $l$ indexed latent variables. We claim there must be exactly $l$ such columns of $\beta$ containing zero entries in every element corresponding to non-leaf nodes, or $\hat{H}$ could not be an optimal solution of Equation (1). We write $\beta$ in block form as

$$\beta = \begin{pmatrix} A & 0 \\ B & C \end{pmatrix},$$

where $A$ is $d - l \times d - l$, $B$ is $d - l \times l$, ad $C$ is $l \times l$. We note that the columns of $\beta$ need not be ordered in this way to achieve our result and that we assume this structure to simplify notation. We derive the inverse of $\beta$ via taking the Schur complement as follows

$$\beta^{-1} = \begin{pmatrix} A^{-1} & 0 \\ -C^{-1}BA^{-1} & C^{-1} \end{pmatrix},$$

where $A^{-1}$ and $C^{-1}$ are full column rank. Now taking, $\hat{Z} = \beta^{-1}Z$, we derive the desired equalities

$$\hat{Z}_i = \begin{cases} A_i^{-1}(Z_0, ..., Z_{d-l}) = linear(Z_{\text{non-leafs}}), & \text{if } Var[J_{\hat{Z}}(\hat{Z}(X))]]_{ii} \neq 0 \\ -C^{-1}BA_i^{-1}(Z_0, ..., Z_{d-l}) + C_i^{-1}(Z_{d-l}, ..., Z_n) = linear(Z) & \text{if } Var[J_{\hat{Z}}(\hat{Z}(X))]]_{ii} = 0, \end{cases}$$

thereby completing the proof. $\qquad\square$

## A.4 Proof of Theorem 1

*Proof.* Assume without loss of generality that the latent variables $Z$ are reverse layerly ordered such that $layer(0) = \{Z_0, ..., Z_{l_0}\}$, $layer(1) = \{Z_{l_0+1}, ..., Z_{l_1}\}$, and so on. Solving for $\hat{H}$ according to the optimization problem framed in Equation (1), it follows from Lemma 3 that

$$\hat{Z}_i = \begin{cases} linear(Z_0, ..., Z_d), & \text{if } Var[J_{\hat{Z}}(\hat{Z}(X))]]_{ii} = 0 \\ linear(Z_{l_0+1}, ..., Z_d), & \text{if } Var[J_{\hat{Z}}(\hat{Z}(X))]]_{ii} \neq 0, \end{cases}$$

where $\{\hat{Z}_i : Var[J_{\hat{Z}}(\hat{Z}(X))]]_{ii} = 0\}$ denotes the representations of $layer(0)$ variables up to upstream layers. Now, if we denote $X'$ as the set of vectors in $\{\hat{Z}_i : Var[J_{\hat{Z}}(\hat{Z}(X))]]_{ii} \neq 0\}$, it follows that

$$X' = H'(Z_{l_0+1}, ..., Z_d),$$

where $H'$ is a full column rank matrix. Thus, viewing $X'$ as our observations of the true non-leaf latent variables, we can utilize Lemma 3 to show that we can recover the $layer(1)$ up to its upstream layer representation. Continuing this method of pruning at each iteration, it is clear to see that we can derive the upstream layer representation for all variables. $\qquad\square$

## A.5 Proof of Theorem 2

*Proof.* Assume there are $n$ distinct layers of $\mathcal{G}$. From Definition 1, it follows that $\hat{\mathcal{E}}_{layer(n)} = \hat{Z}_{layer(n)}$, given $pa(i) = \emptyset$ for all $i \in layer(n)$.

Now, considering the next layer, namely $layer(n-1)$, we can express $\hat{Z}_{layer(n-1)}$ as follows given the principle in Assumption 2,

$$\hat{Z}_{layer(n-1)} = \hat{\mathcal{E}}_{layer(n-1)} + \text{NONLINEAR}(\hat{Z}_{layer(n)}),$$

where $\text{NONLINEAR}(\hat{Z}_{layer(n)})$ denotes the nonlinear relationship between $\hat{Z}_{layer(n-1)}$ and $Z_{pa(layer(n-1))} \in Z_{layer(n)}$ specified by the linear combination of nonlinear functions $\{f_i : i \in layer(n-1)\}$. However, given $\hat{\mathcal{E}}_{layer(n)} = \hat{Z}_{layer(n)}$, $\text{NONLINEAR}(\hat{Z}_{layer(n)}) = \text{NONLINEAR}(\hat{\mathcal{E}}_{layer(n)})$ can be determined. Thus, it follows we can learn $\hat{\mathcal{E}}_{layer(n-1)}$ as the residual of the nonlinear regression of $\hat{Z}_{layer(n-1)}$ on $\hat{\mathcal{E}}_{layer(n)}$ as

$$\hat{\mathcal{E}}_{layer(n-1)} = \hat{Z}_{layer(n-1)} - \text{NONLINEAR}(\hat{\mathcal{E}}_{layer(n)}).$$

Now generalizing to layer $n - i$, we can express $\hat{Z}_{layer(n-i)}$ as

$$\hat{Z}_{layer(n-i)} = \hat{\mathcal{E}}_{layer(n-i)} + \text{NONLINEAR}(\hat{\mathcal{E}}_{layer(n-i+1)}, ..., \hat{\mathcal{E}}_{layer(n)}).$$

Therefore, by the same principle, we can determine $\hat{\mathcal{E}}_{layer(n-i)}$ as the residual of the nonlinear regression of $\hat{Z}_{layer(n-i)}$ on $\hat{\mathcal{E}}_{layer(n-i+1)}, ..., \hat{\mathcal{E}}_{layer(n)}$ as

$$\hat{\mathcal{E}}_{layer(n-i)} = \hat{Z}_{layer(n-i)} - \text{NONLINEAR}(\hat{\mathcal{E}}_{layer(n-i+1)}, ..., \hat{\mathcal{E}}_{layer(n)}).$$

Therefore, we can determine $\hat{\mathcal{E}}_{layer(n-i)}$ for all $i = 0, ..., n$. $\qquad\square$

# B Derivation of Algorithms

We show that the optimization problem in Equation (1) can equivalently be solved by solving the QCQP in Equation (2) for each column sequentially. Given that the $i^{th}$ element of $diag(H^\top J_X(X)H)$ can be expressed as

$$diag(\hat{H}^\top J_X(X)\hat{H})_i = [\hat{H}_i]^\top J_X(X)[\hat{H}_i],$$

we can naturally break our optimization problem into sub-problems of solving for the optimal column vector $[H_i]$ that results in zero variance for the term above. We will now determine an equivalent expression for the variance of this term in terms of a given vector $v \in \mathbb{R}^d$:

$$
\begin{aligned}
Var(v^\top J_X(X)v) &= \mathbb{E}\left[\left(v^\top J_X(X)v\right)^2\right] - \mathbb{E}\left[v^\top J_X(X)v\right]^2 \\
&= \frac{1}{n}\sum_{i=1}^{n}\left(v^\top J_X(x^{(i)})v\right)^2 - \left(\frac{1}{n}\sum_{i=1}^{n}v^\top J_X(x^{(i)})v\right)^2 \\
&= \frac{1}{n}\sum_{i=1}^{n}\left(v^\top J_X(x^{(i)})v\right)^2 - \left(v^\top \bar{J}_X(X)v\right)^2 \\
&= \frac{1}{n}\sum_{i=1}^{n}\left(\left(v^\top J_X(x^{(i)})v\right)^2 - \left(v^\top \bar{J}_X(X)v\right)^2\right) \\
&= \frac{1}{n}\sum_{i=1}^{n}\left(\left(v^\top J_X(x^{(i)})v\right)^2 - 2v^\top J_X(x^{(i)})vv^\top \bar{J}_X(X)v + \left(v^\top \bar{J}_X(X)v\right)^2\right) \\
&\quad + \frac{1}{n}\sum_{i=1}^{n}\left(2v^\top J_X(x^{(i)})vv^\top \bar{J}_X(X)v + 2\left(v^\top \bar{J}_X(X)v\right)^2\right) \\
&= \frac{1}{n}\sum_{i=1}^{n}\left(v^\top J_X(x^{(i)})v - v^\top \bar{J}_X(X)v\right)^2 \\
&\quad + 2\left(v^\top \bar{J}_X(X)v\right)^2 - 2\left(v^\top \bar{J}_X(X)v\right)^2 \\
&= \frac{1}{n}\sum_{i=1}^{n}\left(v^\top (\tilde{J}_X(x^{(i)}))v\right)^2.
\end{aligned}
$$

With this reformulation, the following implication clearly follows:

$$Var(v^\top J_X(X)v) = 0 \iff v^\top \tilde{J}_X(x_i)v = 0 \ \forall i = 1, ..., n. \tag{10}$$

This indicates that the first constraint in the QCQP from Equation (2) solves for a column vector that results in a zero variance term in the diagonal of the estimated Jacobian matrix, as desired. The additional constraints ensure that all of the column vectors added to $\hat{H}$ are linearly independent, fulfilling the full column rank constraint from Equation (1). Thus, continuously solving this QCQP is equivalent to solving the rank-constrained optimization problem from Equation (1).

# C  Details of Experiments

## C.1  Synthetic Data Sampling Procedure

We establish the causal relationships between all nodes and their parents to follow the parametric function $f_i(X) = ||X||^2 + \mathcal{E}_i$, where each $\mathcal{E}_i$ is independently sampled from a mean-zero Gaussian distribution with variance uniformly distributed over [0.1, 1]. For each experiment, we generate random samples of the exogenous noise terms and produce the latent variables via the data generating procedure from Assumption 2. We perform min-max scaling such that every variable is within the range [0, 1]. We perform this scaling, since [33] warns of the fact that valid causal orderings can often be recovered by order of the variables' variance in synthetically generated data, and we wish to show our algorithm performs as desired without this seeming advantage. We randomly sampled $n \times n$ full rank matrices and took the linear transformation of the latent variables on this matrix to derive the corresponding observational samples.

## C.2  Implementation of QCQP Solver

### C.2.1  Perfect Score Estimation

With perfect score estimation, we solve each QCQP to global optimality efficiently using Gurobi optimization solvers [13] on an Apple M2 CPU with 8 cores. We continuously solve the QCQP indicated by Equation (2) with feasibility tolerance set to 0.001, until no feasible solutions remain. We continue by iteratively appending linearly independent column vectors of unit magnitude until $\hat{H}$ is of the desired dimension.

### C.2.2  Score Estimation

When using data-driven score estimation methods, we are not guaranteed to find any column vector that solves the specific QCQP indicated by Equation (2) perfectly. To combat this challenge, we first prune the top 25% of de-meaned Jacobian estimates, $\tilde{J}_X(x)$, by Frobenius norm to remove outliers. We then solve for the minimum value $t$ such that $|v^\top \tilde{J}_X(x^{(m)})v| \leq t$ optimizing over all $v \in \mathbb{R}^d$. Then, we use the same procedure as in the perfect score estimation case with feasibility tolerance set to $t + 0.001$ to solve for the estimated matrix $\hat{H}$.

## C.3  Implementation of Score Estimation

**Stein Estimator.** We implemented the second-order Stein estimator introduced in [5] to generate the point-wise estimates of the score's Jacobian matrices. We use RBF kernels with bandwidth value selected as the median of pairwise distances between points in $X$. Our implementation is adapted from [22] and [35].

**SSM-VR Estimator.** We implemented the sliced score matching with variance reduction (SSM-VR) model developed by [42] to generate functional score estimators. We then estimated the Jacobian of the score estimator using automatic differentiation. Our implementation is adapted from [48].

