# OpenReview forum: "Identifiability Guarantees for Causal Disentanglement from Purely Observational Data"
_NeurIPS.cc/2024/Conference — NeurIPS 2024 poster_

### Official Review · Reviewer_gAcJ · 2024-06-26

**Soundness:** 3
**Presentation:** 3
**Contribution:** 3
**Rating:** 6
**Confidence:** 4

**Summary:**

The authors propose a method to identify causal and exogenous variables in Gaussian Additive Noise Models from purely observational data.

**Strengths:**

The paper proposes a novel approach (the setting might have been largely considered elsewhere; see my comments in **Weaknesses** and **Questions**).


- the proofs (I checked a significant part of the appendix, but not everything) seem to be sound
- the identifiability notion up to upstream layers is an interesting concept (needs to be better discussed what it means and how it compares to other notions)


I chose a conservative score; however, if the authors address my concerns (mostly about making statements more precise) in a satisfying manner, I will increase my score.

**Weaknesses:**

## Major points
- Your statement _our work is the first to establish identifiability guarantees in the purely observational setting without imposing any structural assumptions over the mixing function_ seems to be missing some nuance:
    - assuming an ANM is a structural assumption (and, at least from a theoretical perspective, a rather strong one)
    - you cite [32], which shows that nonlinear ICA can also uncover the causal graph. As I consider nonlinear ICA methods to be purely observational, and as [32] does not assume additivity, this seems to contradict your claim (as, in that case, you would get the DAG and the exogenous variables). Though in [32], there is no level corresponding to $H$ -- if this is the novely, then please be more precise.
- optimizing the Jacobian is computationally very expensive; I haven't found any discussion on this aspect
- a more detailed discussion on delineating the contribution would be highly suggested (from both score-based methods and [32])
- I also encourage the authors to address the (limitations of) identifiability up to upstream layers; though, as I wrote above, find the concept interesting, my understanding is that this is a very limited identifiability notion



## Minor points
- line 93: "principal" -> "principle"
- use different $C, P_\pi$ for Defs 1 and 2
- the definition of, e.g., $\beta$ in Lem 1 comes a bit late; this makes the reader wonder what it stands for
- resolve each abbreviation and symbol in each Table and Figure caption, even if you defined those quantities in the main text

**Questions:**

- What exactly do you mean by asymmetries? I only found the related line 46, and that does not seem to be any novel key insight, so I am assuming I am missing your point.
- What do you exactly mean by causal disentanglement?
- Why do you consider Gaussian exogenous variables when that causes the (unsurprising) impossibility result in 3.3
- Can you use insights from score-matching-based methods to improve your identifiability results (those methods can easier identify leaf nodes, whereas your method works better for root nodes; I reckon you draw some connections, but I could not find this specific aspect)
- In Fig. 4, what is the MAC for SSM and STEIN? Am I interpreting it correctly that it can be everything?
- How does identifiability up to layers (particularly, distinguishing leaf and non-leaf variables) relate to block-identifiability (if it does relate), or, particularly, to content and style latent variables (cf. [1])


## References
- [1] Von Kügelgen, J., Sharma, Y., Gresele, L., Brendel, W., Schölkopf, B., Besserve, M., & Locatello, F. (2021). Self-supervised learning with data augmentations provably isolates content from style. Advances in neural information processing systems, 34, 16451-16467.

**Limitations:**

The authors addressed broader impact. I asked for clarifications regarding computational and novelty aspects.

---

> ### Author Rebuttal · Authors · 2024-08-06
>
> Thank you for your detailed review! We appreciate that you found our proofs sound and our newly defined notion of identifiability to be interesting. We would like to address your comments below:
>
> > **“What do you exactly mean by causal disentanglement?”**
>
> Following this review (https://arxiv.org/abs/2206.15475), we define causal disentanglement to be the process of learning about latent variables $Z$ from the set of observational variables $X$, such that $X = g(Z)$ for some mapping $g$, and $Z$ factorizes as $p(Z) = \prod_{i=1}^n p(Z_i | Z_{pa(i)}, \epsilon_i)$, where $\epsilon_i$ is an exogenous noise term associated with $Z_i$. We additionally note that this problem has also been called “causal representation learning” in the literature. We will refine Section 2 to make this more clear.
>
> > **“you cite [32] …  there is no level corresponding to H -- if this is the novely, then please be more precise.”**
>
> > **“a more detailed discussion on delineating the contribution ...”**
>
> Thank you for your suggestion. We would like to clarify that prior score-based methods (including [32]) consider the problem of causal structure learning when all causal variables are _fully observed_, while we consider the more difficult problem of causal disentanglement, where the causal variables are _inherently latent and can only be viewed through an unknown mixing function_, which we denote by $H$. Using the notation defined in our paper, [32] would assume direct access to the latent variables, $Z$, for which we derive methods to learn representations of. We will add a more thorough discussion in Section 1.1 indicating this distinction between our contributions.
>
> > **“assuming an ANM is a structural assumption…”**
>
> We wanted to distinguish structural assumptions from functional assumptions as restrictions to _just the mapping from the latent to the observed space_ as defined by the mixing function. For clarity, previous works in purely observational causal disentanglement rely on structural assumptions on this mapping, the most common one being the pure child assumption ([11,14,19,52,53] in the manuscript), assuming that each latent variable has multiple observational variables for which they are the only parent.  We in contrast allow any observational variable to be determined by any subset of latent variables. We briefly discussed this in lines 75-81, but we will revise this discussion to make it more clear.
>
> > **“optimizing the Jacobian is computationally very expensive; I haven't found ...”**
>
> Thank you for this comment. We acknowledge that optimizing the Jacobian in Eq (1) can be computationally laborious. We briefly discussed this difficulty in lines 217-220 in Section 4.1, which is followed by our proposed solution. We also discussed the difficulty of estimating the Jacobian in lines 275-281 in Section 5.1. We will add pointers to these places in the introduction.
>
> > **“..encourage the authors to address the (limitations of) identifiability up to upstream layers …”**
>
> > **“Can you use insights … to improve your identifiability…”**
>
> Thank you for your suggestion. The primary limitation of identifiability up to upstream layers is that variables which are more downstream in the underlying causal graph have weaker identifiability guarantees. Conversely, variables that are more upstream in the causal graph, or more influential in other words, have stronger identifiability guarantees. As shown in Section 3.3, these identifiability results can _not_ be improved without further assumptions. It is currently unclear what minimal additional assumptions are required to achieve stronger identifiability guarantees. We note this as an important question for future work, and we will explicitly add this discussion to Section 6.
>
> > **“What exactly do you mean by asymmetries?”**
>
> Asymmetries refers to the asymmetric relationships between causes and effects present in this model. More specifically, we will have Gaussian residuals if we regress the effect over causes, but not the other way around. These relationships are useful in establishing causal directions, which we utilize to learn the causal layers of the underlying graph.
>
> > **“Why do you consider Gaussian exogenous variables when that causes …”**
>
> The Gaussian noise assumption is common for nonlinear additive noise models given their nice theoretical properties that allow for  the whole causal graph to be learned when all causal variables are observed (c.f., [33] referred in the manuscript). Our theoretical proofs similarly rely on this Gaussianity assumption, which is why we consider them even though they cause the specified impossibility result. Considering whether stronger identifiability results could be achieved under non-Gaussianity assumptions is an interesting direction for future work.
>
> > **“In Fig. 4, what is the MAC for SSM and STEIN? …”**
>
> The text “SSM” and “STEIN” in Figure 4 are intended to be labels of the black and red vertical lines indicating these methods’ average signal-to-error ratio when used to estimate the pointwise Jacobians of the score, which is 6 and 2 dB respectively. We will make these labels more clear with a legend in our revision. The MAC of the exogenous noise estimates using these estimation methods for various sample sizes is depicted in Table 2.
>
> > **“… relate to block… content and style latent variables”**
>
> Since identifiability up to layers means that each causal variable can be identified up to transformation of itself and all variables in or above its layer, this means that we can achieve block-identifiability of each variable and all variables in or above its layer. In particular, this means the root variables can be identified up to their own block. For content and style variables, as content variables are in the layers above style variables, they can be identified up to their own block.
>
>
> ---
> Thank you for the minor points as well. We will revise accordingly.

---

> > ### Comment · Reviewer_gAcJ · 2024-08-07
> > **Score change 4->5**
> >
> > Thank you for your detailed response!
> >
> > - I'd consider using causal representation learning, as it seems to me to be a more accepted and precise notion (disentanglement in general representation learning is usually a vague concept)
> > - I'd consider using "graphical" instead of "structural." As another reviewer pointed out, using "structural" is presumably very confusing. Nonetheless, assuming ANMs is still a strong assumption

---

> ### Author Response · Authors · 2024-08-08
> **Response to comment**
>
> Thank you for the discussion and for updating the scores!
>
> Thanks for the suggestion! We will revise the manuscript to clarify the concepts regarding (1) causal representation learning and causal disentanglement, and (2) structural restrictions and graphical restrictions. Our usage and initial thoughts aim to align with recent prior works in this area (e.g., the usage of causal disentanglement in [1], and structural restrictions in [2]). Nonetheless, we will clarify these concepts to improve readability.
>
> In addition, we would like to comment on why we believe that while ANM poses certain assumptions, it can still be a useful model in many scenarios:
>
> **The Additive Noise Model Assumption**
>
> When compared to alternative assumptions (e.g., independent components, which are a special case of ANM) in identifiable representation learning, we believe that ANM is relatively less restrictive, as it allows dependencies between the latent components. These dependencies are important, in our opinion, as the latent factors should be related to each other. Additionally, in the causal structure learning setting where all causal variables are observed, ANMs are frequently used because (1) their theoretical properties are well understood, (2) efficient methods exist for learning them, and (3) they can fit non-parametric relationships, making them flexible for modeling many real-world causal systems, such as gene regulatory networks [3]. Therefore we believe that it would be helpful to devise understandings and methods to extend these models to the causal representation learning setting, where we can handle perceptual and non-tabular observations.
>
> ---
> References:
>
> [1] Causal Machine Learning: A Survey and Open Problems \
> [2] Linear causal disentanglement via interventions \
> [3] Estimation of genetic networks and functional structures between genes by using Bayesian networks and nonparametric regression

---

> > ### Comment · Reviewer_gAcJ · 2024-08-08
> > **Score re-evaluation**
> >
> > Thanks for the detailed response. I went through your earlier response again, and in this light, I modified my score to 6.

---

> > > ### Author Response · Authors · 2024-08-08
> > > **Response**
> > >
> > > Thank you again for the discussion! We really appreciate the suggestions.

---

### Official Review · Reviewer_2B2i · 2024-07-12

**Soundness:** 2
**Presentation:** 2
**Contribution:** 1
**Rating:** 4
**Confidence:** 4

**Summary:**

This paper concerns itself with the problem of causal disentanglement from purely observational data. The setting is that data X is generated as X = H.Z where H is a linear matrix and Z are latent variables that follow a nonlinear additive Gaussian noise model. In this setting, it is shown that the latent variables can be identified upto upstream layers (defn 1), when given access to infinite data.

The idea is to compute the score functions (gradient of log-likelihood) and use their structural properties to recover the layers. This is then converted into an algorithm, which computes the desired score functions, using standard score-estimation methods and an application of quadratic programming to identify the model. The technique is validated numerically via simulated experiments.

This setting is similar to [1] however interventional data is assumed to not be available, however, the identifiability results seem weaker as expected. However, the claims made in the paper seem too strong (e.g. this is not the first paper to consider the purely observational setting, see [2]) and the proof techniques also seem to have appeared in prior works such as [3] (see weaknesses).

### References:

- [1] Score-based causal representation learning with interventions. B. Varici, E. Acarturk, K. Shanmugam, A. Kumar, A. Tajer.

- [2] Identifiability of deep generative models without auxiliary information. B Kivva, G Rajendran, P Ravikumar, B Aragam

- [3] Score matching enables causal discovery of nonlinear additive noise models. P Rolland, V Cevher, M Kleindessner, C Russell, D Janzing, B Schölkopf, F Locatello

**Strengths:**

- Compared to a flurry of recent works that assume existence of interventional data to recover latent variables in data, this work studies purely observational data, which is a more useful setting.

- The notion of identifibility upto layer-wise transformations is novel to the best of my knowledge and could potentially be useful elsewhere.

**Weaknesses:**

- L184/L284 claims there are no structural results on the mixing function, however assumption 1 assumes that the mixing function is linear, which is a pretty big restriction. This weakens the claim of this paper.

- L81-82 says "our work is the first to establish identifiability guarantees in the purely observational setting without imposing any structural assumptions over the mixing function". However, the prior work [2] also considers the purely observational setting and the assumption they make on the mixing function is piecewise-linearity, which is much weaker than the linearity assumption made here. That makes this claim not valid, could the authors clarify this point?

- This work seems to be an extension of [3] where an additional linearity is added. The lemmas and proofs also seem similar. Could the authors describe the main differences in the assumptions and/or additional difficulties in the proof?

- Experiments are on synthetic data. While they weakly validate the theoretical results, an application to real-life would be needed to see if these ideas extend to practice.

**Questions:**

Some questions were raised above.

- Identifiability up to layers seems a bit hard to grasp. While theoretically true, what does it mean for the practitioner?

- Typo in L26: extend -> extent

**Limitations:**

Limitations have been discussed.

---

> ### Author Rebuttal · Authors · 2024-08-06
>
> Thank you for appreciating our problem setting and for recognizing the novelty of our defined notion of identifiability. We would like to address your concerns and questions below:
>
> > **“L184/L284 claims there are no structural results on the mixing function, however assumption 1 assumes that the mixing function is linear … ”**
>
> To clarify, we consider the assumption of linear mixing between the latent and observed variables to be a _functional_ assumption rather than a _structural_ restriction, which refers to hard limits on the sets of latent variables that can determine each observational variable. In our setting, we allow any subset of latent variables to determine any observational variable. This is in contrast to previous works in purely observational causal disentanglement, which rely on the pure child assumption ([11,14,19,52,53] in the manuscript), assuming that each latent variable has multiple observational variables for which they are the only parent.
>
> We note that we chose to assume a linear mixing as it is essential to the proofs of our theoretical guarantees. However, our results additionally hold when the true mixing function can be reduced to linear mixing, such as in the case of polynomials being reduced to linear mappings (c.f., [2,57] in the manuscript).
>
> > **“L81-82 says "our work is the first to establish identifiability guarantees in the purely observational setting without imposing any structural assumptions over the mixing function". However, the prior work [2] …”**
>
> Thank you for raising this concern. The setting considered in [2] is very different from the problem setting in our paper. While [2] considers the purely observational setting, their results are dependent on the assumption that the latent variables are distributed according to a Gaussian mixture model, which does not specifically model the causal graph, and does not encapsulate the nonlinear additive Gaussian noise model that we consider. Therefore, the results in [2] are not transferable to our setting, and our original claim in L81-82 holds since we consider the setting of causal disentanglement. We will refine the claim in lines 81-82 to clarify this.
>
> > **“This work seems to be an extension of [3] where an additional linearity is added… describe the main differences in the assumptions and/or additional difficulties in the proof?”**
>
> You are correct that our work is an extension of [3]. However, the important distinction between our works is that [3] considers the setting where the causal variables are fully observed, while we consider the more difficult setting where the causal variables are latent and can only be viewed through an unknown mixing function, which is where the additional difficulties lie.
>
> The proofs in [3] utilize variance properties on the diagonal elements of the Jacobian over the score of the causal variables to derive a topological ordering. While we utilize this result, it is not sufficient given we don't have access to the specified causal variables. In Lemma 1, we prove that we can only estimate this desired Jacobian up to an unknown quadratic parameterized by the matrix $\beta$, where $J_{\hat{Z}} (\hat{z}) = \beta^{T} J_Z(z) \beta$. We therefore must derive additional information about the whole Jacobian, not just the diagonal as in [3]. In Lemma 2, we derive properties of the estimator for the unknown matrix $\beta$ such that $J_{\hat{Z}} (\hat{z})$ has zero variance terms in its diagonal. We then take it a step further in Lemma 3 and Theorem 1, to demonstrate how these properties on $\beta$ allow us to derive layer-wise representations, by maximizing the number of zero-variance terms in the diagonal of the Jacobian. This gives rise to a simple principle in Eq (1) to achieve identifiability, which serves as foundation of our algorithm.
>
> > **“Experiments are on synthetic data .. an application to real-life would be needed to see if these ideas extend to practice.”**
>
> We thank the reviewer for this comment. Similar to [46,18,9] in our manuscript, we view this paper as having a primarily theoretical contribution, in which our experiments on synthetic data provide a convincing proof-of-concept for our main results.
>
> We acknowledge the importance of real-world experiments and do believe there are many real-world scenarios, such as topic modeling in natural language for example, that could be interesting settings to evaluate our methods. In particular, the method could be used to identify hierarchical topics at different layers of the underlying causal structure. Such experiments would however necessitate an extensive amount of additional thought in experimental set up, which we believe is out of the scope of our current work, which mainly focuses on theoretical guarantees.
>
> > **“Identifiability up to layers seems a bit hard to grasp… what does it mean for the practitioner?”**
>
> For causal systems which have an inherent hierarchical structure, upstream layer representations will capture all of the information used in determining each particular level of variables. This can be best explained with an example. Consider we wish to disentangle information about a latent genealogical tree, which we don’t have the ability to intervene on as it is no longer active. Intuitively, each layer representation would contain all of the prior ancestral information used to determine a given generation's traits, where the top layer would only contain information about the original ancestors and the bottom layer representation would contain all hereditary information. The additional layer-wise noise representations would capture each generation’s level of exogenous noise, providing an understanding of how traits could have spawned over time.
> We will add this particular example to the introduction to build intuition for how our work can be useful beyond just the theoretical guarantees.
>
>
> > **“Typo in L26: extend -> extent”**
>
> Thank you for identifying this. We will revise accordingly.

---

> ### Comment · Reviewer_2B2i · 2024-08-11
> **Score update**
>
> Thank you for your response. I understand the author’s clarifications on the differences between the works that I have cited, but my concern is more with the framing of the contributions and it still seems like overselling of the work even if there are clear technical differences from prior works. While there is some intuition behind identifiability upto layers that the authors give, I still find it hard to understand why it would be useful either theoretically or in practice. Thanks for the other technical clarifications as well, I have increased my score.

---

> > ### Author Response · Authors · 2024-08-12
> > **Response to further points**
> >
> > Thank you for the response and the further comments. We are glad that we are able to make the technical clarifications.
> >
> > Regarding the remaining concerns about the framing of the contributions, we really appreciate the comments and will try to clarify it via the following changes:
> > 1. Causal disentanglement in the purely observational setting: in the related work section on causal disentanglement (line 81-82), we stated that “our work is the first to establish identifiability guarantees in the purely observational setting without imposing any structural assumptions over the mixing function”. We will modify this claim to “our work establishes identifiability guarantees of causal disentanglement in the purely observational setting, without imposing any structural assumptions over the mixing function”. In addition, we will add a discussion paragraph to clarify that identifiability of latent factors in the purely observational setting has been considered outside of causal disentanglement, such as in the work [2] mentioned by the reviewer.
> > 2. Score-based approaches in causal discovery vs. causal disentanglement: in the related work section on “score matching in causal discovery” (line 91-92), we commented that “extending these ideas to causal disentanglement is difficult, since we do not observe the latent factors and can only estimate the log-likelihood of the observed variables”. We will add in the technical differences of our proof versus the original proof in the causal discovery setting. In particular, we will add a pointer to section 3.2 (which is where our main lemmas and theorems sit), in which we will incorporate our technical clarification in the earlier response: “The proofs in [3] utilize variance properties on the diagonal elements of the Jacobian over the score of the causal variables to derive a topological ordering. While we utilize this result, it is not sufficient given we don't have access to the specified causal variables. In Lemma 1, we prove that we can only estimate this desired Jacobian up to an unknown quadratic parameterized by the matrix , where …”
> >
> > We hope that these will be sufficient to clarify our contributions and avoid overselling it.
> >
> > Thank you for the other comments as well. In response to it, we would like to give our perspectives regarding why we believe identifiability up to layers might still be useful albeit its limitations:
> > in the emerging field of causal disentanglement, full identifiability of latent causal model is not possible without additional assumptions, which is why many works proposed to consider, e.g., structural or sparsity restrictions on the mixing function, or access to single-node interventions. However, as we tried to illustrate in lines 39-44, these assumptions might be limiting and impractical in many settings, which is why we choose to step back and study what can be identified without interventions or structural restrictions. We choose to study nonlinear additive noise models, as they inherit the nice theoretical properties in the causal discovery setting and allow modeling of non-parametric causal mechanisms. In this case, we attain a full theoretical understanding of what can learned, by showing partial identifiability of up to causal layers, which cannot be improved without additional assumptions. Practically, this would mean that more upstream variables in a hierarchical causal structural can be disentangled easier. For example, in the context-style model in [4], our results show that the context variable can be identified up to themselves.
> >
> > ---
> > References:
> >
> > [4]  Self-supervised learning with data augmentations provably isolates content from style.

---

### Official Review · Reviewer_1YMg · 2024-07-13

**Soundness:** 3
**Presentation:** 4
**Contribution:** 3
**Rating:** 6
**Confidence:** 3

**Summary:**

This paper studies the identifiability issue of causal disentanglement from observational data, within the setting of nonlinear causal model with additive Gaussian noise and linear mixing. An interesting result is that the causal variables can be identifiable at most up to a layer-wise transformation, based on a recent score-based causal discovery method. A practical algorithm is then proposed.

**Strengths:**

- The paper reconsiders the fundamentals of an import problem, which is very meaningful.

- The theory, in particular the concept of layer-wise identifiability, in new and motivating.

- The proposed method is practical.

- The paper is well-written and it is enjoyable to have a read.

**Weaknesses:**

A key motivation of the paper relies on the reasonability of assuming interventions on latent factors. Then this paper assumes non-linear function with Gaussian additive noise and linear mixing, but these assumptions cannot be tested from observations, either. In other words, these assumptions are, in my view, alternative assumptions on the same problem, but do not relax the previous assumptions. Besides, in this setting, the dimension of latent factors is assumed to be known, which may be a limitation.

**Questions:**

Overall, the paper studies a very interesting and meaningful problem. Within the considered setting, quite some interesting results are obtained. My questions are:

1. As mentioned in the weakness part, this paper assumes non-linear function with Gaussian additive noise and linear mixing, but these assumptions cannot be tested from observations, either. In other words, these assumptions are, in my view, alternative assumptions on the same problem, but do not relax the previous assumptions. Besides, in this setting, the dimension of latent factors is assumed to be known, which may be a limitation. Can you clarify on this?
2. the key part of identifiability theory seems to rely on the recent technique of score-based causal discovery method. In your setting, the latent factors themselves are identifiable before the linear mixing. I am wondering if other forms identifiable SCMs for the latent variables can also be identified?
3. Regarding Proposition 1.: while a counter example is sufficient for this theorem, I'd like to ask if this failure case can be avoided by putting additional assumptions? e.g., what if we put a technical condition that $a_1b_1\sigma_1^2+a_2b_2\sigma_2^2\neq 0$ or that $a_1, b_1, a_2, b_2>0$.?
4. typo: line 26: extend -> extent

**Limitations:**

See above.

---

> ### Author Rebuttal · Authors · 2024-08-06
>
> Thank you for your encouraging review! We appreciate you recognizing both the importance of our work and the merit of our theoretical findings and algorithmic approach. We would like to address your additional comments and questions below:
>
> > **“…this paper assumed non-linear function with Gaussian additive noise and linear mixing, but these assumptions cannot be tested from observations, either. In other words, these assumptions are, in my view alternative assumptions on the same problem, but do not relax the previous assumptions.”**
>
> We thank the reviewer for this comment, and we take this opportunity to give more intuition on our model assumptions and clarify the difference between them and assumptions on interventions. Although it may seem like a strong assumption to consider a non-linear additive Gaussian noise model as the representation of the underlying causal network, we note that this model type is frequently assumed in causal inference given their theoretical properties are well understood and there exist many methods to learn their structure in the fully observable setting [1-4]. Additionally, these models are known to be more flexible than linear additive noise models given their ability to fit non-parametric relationships, making them a commonly assumed model for many real-world causal systems such as gene regulatory networks [5] without needed verification. For the mixing function, we chose to assume a linear mixing as it is essential to the proofs of our theoretical guarantees. However, our results also hold when the true mixing function can be reduced to linear mixing using existing techniques, such as in the case of polynomials being reduced to linear mappings (c.f. [6,8]). It would be interesting in future work to understand to what other settings our results can be extended.
>
> We recognize that there are different levels of assumptions made on the latent model and the mixing function, but we do not consider access to interventions as alternative assumptions on the same problem. While our assumptions are restricting _the model class of the data-generating process_, the assumption of interventions assume _existence of additional data / environments_, which is intrinsically different. This is also why we separate and highlight the assumption on “_data_” from “_latent model_” and “_structural mixing_” in Table 1 of our paper. However, we want to clarify that we do not consider our work as a strict relaxation of prior works, as the works which assume interventions usually compare multiple environments and might arrive at stronger identifiability results by imposing assumptions on the number / types of interventions.
>
>
> > **“Besides, in this setting, the dimension of latent factors is assumed to be known, which may be a limitation.”**
>
> Thank you for this comment. We would like to clarify that we do _not_ assume the dimension of the latent vector is known. Given we consider linear mixing, we are able to solve for the latent dimension $n$ by solving for the smallest integer $\hat{n}$ such that there exists a full column rank matrix $\tilde{H} \in \mathbb{R}^{d \times \hat{n}}$ where $\tilde{Z}= \tilde{H}^{\dagger} X$ has an open support in $ \mathbb{R}^{\hat{n}}$, similar to Lemma 1 in [6]. We will add this information to Section 2 for clarity.
>
> > **“the key part of identifiability theory seems to rely on the recent technique of score-based causal discovery method. In your setting, the latent factors themselves are identifiable before the linear mixing. I am wondering if other forms identifiable SCMs for the latent variables can also be identified?”**
>
> In considering other works that utilize score-matching to identify SCMs in the fully observable setting, we believe that our theoretical result could be extended to learn the upstream layer representations of nonlinear additive models with generic noise as an extension of [7], by modifying the principal to achieve identifiability in Eq (1) of our Lemma 3 to accommodate generic noise. The practical algorithm needs to be adapted accordingly as well. We will add this discussion to the paper.
>
> > **“Regarding Proposition 1.: while a counter example is sufficient for this theorem, I'd like to ask if this failure case can be avoided by putting additional assumptions? e.g., …”**
>
> We appreciate this question regarding conditions for further identifiability. With additional faithfulness assumptions on the data generating process, we can potentially further identify the latent variables beyond "up to upstream layers". For example, assuming a generalized notion of faithfulness and access to soft interventions, [6] demonstrates that the underlying causal graph can be identified up to transitive closure. The mentioned condition by the reviewer can serve as one such faithfulness assumption for the 2-variable scenario. However, it is currently unclear what minimum faithfulness assumptions are required to achieve stronger identifiability guarantees in general scenarios, which we view as an important question for future work.
>
> > **“typo: line 26: extend -> extent”**
>
> Thank you for identifying this. We will revise accordingly.
>
> ---
> References:
>
> [1] Nonlinear causal discovery with additive noise models \
> [2] Causal discovery with continuous additive noise models \
> [3] Score matching enables causal discovery of nonlinear additive noise models \
> [4] CAM: Causal additive models, high-dimensional order search and penalized regression \
> [5] Estimation of genetic networks and functional structures between genes by using Bayesian networks and nonparametric regression \
> [6] Identifiability guarantees for causal disentanglement from soft interventions \
> [7] Causal discovery with score matching on additive models with arbitrary noise \
> [8] Interventional causal representation learning

---

> > ### Comment · Reviewer_1YMg · 2024-08-08
> >
> > Thanks for your reponse. I main my score.

---

> > > ### Author Response · Authors · 2024-08-08
> > > **Response**
> > >
> > > Thank you again for the suggestion and the discussion!

---

### Official Review · Reviewer_zdJm · 2024-07-17

**Soundness:** 2
**Presentation:** 2
**Contribution:** 3
**Rating:** 6
**Confidence:** 3

**Summary:**

This paper investigates causal disentanglement, learning latent causal factors from observational data without interventions. It identifies latent factors in nonlinear causal models with additive Gaussian noise and linear mixing, showing that causal variables can be identified up to a layer-wise transformation. The authors propose an algorithm based on quadratic programming over score estimation and validate it with simulations, demonstrating meaningful causal representations from observational data.

**Strengths:**

1. The paper presents a method to identify causal variables from observational data without interventions.
2. It also provides the theoretical analysis, and demonstrates that latent variables can be identified up to a layer-wise transformation consistent with the underlying causal ordering, with no further disentanglement possible.

**Weaknesses:**

1. The method proposed in this paper relaxes some assumptions, making it potentially more applicable to real-world scenarios. It would be better to provide the performance of applying the proposed method to real-world data.

2. In line 475, the denominator of the first term after the third equals sign in the equation should be differentiated with respect to  $z_l$ instead of $k_l$?
 .

**Questions:**

See the weaknesses above.

**Limitations:**

Not applicable.

---

> ### Author Rebuttal · Authors · 2024-08-06
>
> Thank you for recognizing the strength of our theoretical analysis and our proposed method! We would like to address your comments below:
>
> > **“The method proposed in this paper relaxes some assumptions, making it potentially more applicable to real-world scenarios. It would be better to provide the performance of applying the proposed method to real-world data.”**
>
> We thank the reviewer for this suggestion. Similar to [1,2,3], we view this paper as having a primarily theoretical contribution, where our experiments on synthetic data provide a proof-of-concept for our main results. While we acknowledge the importance of real-world experiments, a major challenge lies in the fact that ground truth latent variables are not specifically labeled in many real-world datasets, making it difficult to evaluate the precise accuracy of estimated latent variables. Similar constraints are present in many previous works in causal disentanglement, where evaluations are generally based on synthetic data.
>
> However, we do believe there are many real-world scenarios, such as topic modeling in natural language for example, that could be interesting settings to evaluate our methods. In particular, the method can be used to identify hierarchical topics at different layers of the underlying causal structure. Another example is learning a latent genealogical tree, where each layer representation would contain all of the prior ancestral information used to determine a given generation's traits. Such experiments would however necessitate an extensive amount of additional thought in experimental set up, which we believe is out of the scope of our current work, which mainly focuses on theoretical guarantees. We however recognize the importance of real-world applications and will add these examples to the introduction to build intuition of how our work might be useful beyond the theoretical guarantees.
>
> > **In line 475, the denominator of the first term after the third equals sign in the equation should be differentiated with respect to $z_l$ instead of $k_l$?**
>
> Thank you for pointing this out. This term should be differentiated with respect to $z_l$ instead of $k_l$. We will edit accordingly.
>
> ---
> References:
>
> [1] General identifiability and achievability for causal representation learning \
> [2] Learning latent causal graphs via mixture oracles \
> [3] Learning linear causal representations from Interventions under general nonlinear mixing

---

> > ### Comment · Reviewer_zdJm · 2024-08-09
> >
> > Thank you for the clarifications. I’m happy to increase my evaluation score.

---

> > > ### Author Response · Authors · 2024-08-09
> > >
> > > Thank you for the discussion and for updating the score! We are grateful to the suggestions.

---

### Decision · Program_Chairs · 2024-09-25

**Decision:**

Accept (poster)

**Comment:**

The submission provides a novel identifiability result for the challenging case of causal representation learning based on observational data only. Reviewers have overall appreciated the significance of the contribution and the insights it provides with the concept of layerwise identifiability. The AC recommends acceptance.